# Identifying Key Regulatory Genes in Drug Resistance Acquisition: Modeling Pseudotime Trajectories of Breast Cancer Single-Cell Transcriptome

**DOI:** 10.3390/cancers16101884

**Published:** 2024-05-15

**Authors:** Keita Iida, Mariko Okada

**Affiliations:** Institute for Protein Research, Osaka University, Suita 565-0871, Osaka, Japan; mokada@protein.osaka-u.ac.jp

**Keywords:** single-cell transcriptome, pseudotime analysis, nonautonomous ordinary differential equation, drug resistance, breast cancer, intracellular signaling

## Abstract

**Simple Summary:**

Despite recent advancements in experimental technology for genome-wide molecular profiling, our understanding of the dynamic mechanism underlying cancer drug resistance remains limited. In this study, we present an approach that combines mathematical modeling with the pseudotime analysis of single-cell time-series transcriptome data of drug-treated breast cancer cells. Our method identifies approximately 600 genes out of 6000 exhibiting multistable expression states, including *RPS6KB1*, a predictor of poor prognosis, cell survival, and growth in estrogen-receptor-positive breast cancers. The bifurcation analysis elucidates the regulatory mechanisms of the key regulatory genes, which can also be mapped into a molecular network based on cell survival and metastasis-related pathways, providing a comprehensive understanding of the interplay between signaling pathways and regulatory genes. Our method serves as a powerful tool for deciphering the complexities of drug resistance mechanisms in human diseases.

**Abstract:**

Single-cell RNA-sequencing (scRNA-seq) technology has provided significant insights into cancer drug resistance at the single-cell level. However, understanding dynamic cell transitions at the molecular systems level remains limited, requiring a systems biology approach. We present an approach that combines mathematical modeling with a pseudotime analysis using time-series scRNA-seq data obtained from the breast cancer cell line MCF-7 treated with tamoxifen. Our single-cell analysis identified five distinct subpopulations, including tamoxifen-sensitive and -resistant groups. Using a single-gene mathematical model, we discovered approximately 560–680 genes out of 6000 exhibiting multistable expression states in each subpopulation, including key estrogen-receptor-positive breast cancer cell survival genes, such as *RPS6KB1*. A bifurcation analysis elucidated their regulatory mechanisms, and we mapped these genes into a molecular network associated with cell survival and metastasis-related pathways. Our modeling approach comprehensively identifies key regulatory genes for drug resistance acquisition, enhancing our understanding of potential drug targets in breast cancer.

## 1. Introduction

Breast cancer has been the most prevalent cancer among women and a leading cause of death since the late 20th century [1,2,3]. In the United States, approximately 75% of breast cancers are diagnosed as luminal A subtype [4], indicating their dependency on the estrogen receptor (ER) for growth. Tamoxifen, a widely used endocrine therapy for ER-positive breast cancer, has resulted in improvements in overall survival [5,6]. However, a significant proportion of patients treated with tamoxifen experience a relapse within 5–15 years, often resulting in metastasis and mortality [5,6]. Despite accumulating knowledge on the genetic and nongenetic molecular mechanisms of endocrine resistance, such as *ESR1* mutation, transcriptomic and epigenetic changes, and the activation of signaling pathways [6,7,8], our understanding of the dynamic processes remains limited, requiring a comprehensive and systemic approach.

Recently, single-cell RNA-sequencing (scRNA-seq) analysis using MCF-7 cells, a luminal A subtype breast cancer model, has provided insights into drug resistance acquisition, revealing cellular heterogeneity and dynamic cell transitions. Notably, a “pre-adapted” subpopulation exhibiting the plasticity between drug tolerance and resistance has been identified in MCF-7 cells [9,10]. These studies investigated the activation of cell survival signals, such as the NF-κB pathway. Furthermore, Magi et al. reported altered metabolic regulation, cell adhesion, and histone modification based on the analysis of time-series scRNA-seq data under tamoxifen treatment [11]. While these studies have uncovered important genes and pathways associated with drug resistance acquisition, gaps still exist in our understanding of regulatory mechanisms at the systems level. Hence, a further exploration for potential drug targets is warranted.

To address these issues, we propose an approach that combines mathematical modeling with a pseudotime analysis using time-series transcriptome data, which automatically identifies key regulatory genes. We first identified potential drug-sensitive and -resistant cells by analyzing the time-series scRNA-seq data of tamoxifen-treated MCF-7 cells. Next, we performed a pseudotime analysis, from which we observed a transcriptional relay involved in cAMP, PI3K-AKT-mTOR, RHO-GTP, NF-κB, and NOTCH pathways. We identified genes exhibiting multistable expression states by modeling the individual gene expression patterns and conducting a steady-state analysis. Finally, we propose a transcriptional relay network model that includes cell survival, focal adhesion, and epithelial–mesenchymal transition (EMT), along with the key regulatory genes. Our results suggest that the modeling approach through a pseudotime analysis of the single-cell transcriptome serves as a powerful tool for understanding the drug-induced cellular transition.

## 2. Results

### 2.1. Overview of the Analysis Workflow

To theoretically address the challenge of identifying effective targets for regulating complex molecular systems during drug resistance acquisition (Figure 1a), we developed an approach combining mathematical modeling and a pseudotime analysis using time-series transcriptome data, which automatically identifies key regulatory genes (Figure 1b). Previous research on cell differentiation revealed that transcription factor networks exhibiting multistable states simulate dynamic cell transitions of progenitor cells into different terminal states [12,13,14]. This suggests that genes with multistable expression states could serve as potential indicators of cell differentiation, potentially regulated by switch-like responding pathways. To identify these key regulatory genes, we modeled each gene expression using an ordinary differential equation, determining whether the expression state in the long-time limit is mono- or multistable. Although multistability was investigated in the steady state, we considered these “multistable genes” key regulators, assuming the cell fate is determined within the long-time limit. Subsequently, as discussed later, we constructed a transcriptional relay network model for cell survival, focal adhesion, and EMT.

### 2.2. Dissecting the Heterogeneity and Defining Pseudotime for MCF-7 Cells

To understand transcriptional regulation dynamics underlying drug resistance acquisition, we analyzed a published dataset containing time-series scRNA-seq data of MCF-7 cells subjected to continuous tamoxifen treatment [11]. The data were collected at weeks 0 (before tamoxifen treatment), 3, 6, and 9, referred to as W0, W3, W6, and W9, respectively (Figure 2a). The original authors reported that cell growth was significantly inhibited by the fifth week, after which two resistant subpopulations emerged: one displaying altered metabolic regulation and the other exhibiting cell adhesion and histone modification. To provide a deeper insight into these findings, we first analyzed the W0 data to identify potential drug-sensitive and -resistant cells present before tamoxifen treatment. Subsequently, we analyzed the W0–W9 data, allowing us to derive the pseudotime gene expression profiles.

After normalizing the W0–W9 data (Appendix A and Appendix B), we initially focused on W0. Using Seurat (v4.4.0) [15], a standard software for scRNA-seq data analysis, we clustered the cells into four distinct subpopulations (W0-I, -II, -III, and -IV) and inferred the cell cycle phases (Figure 2b). We examined differentially expressed genes to assess potential drug-resistant states (Figure 2c). We observed the significant upregulation of *IGFBP5* and *JUN* in the W0-I subpopulation. Previous studies have implicated IGFBP5 in cancer inhibition by temporally inactivating the insulin-like growth factor receptor [16], and JUN has been linked to increased tamoxifen sensitivity through protein kinase C activation [17,18]. Conversely, a positive cell cycle regulator *CCND1* [19] and apoptosis inhibitor *XBP1* [20] were significantly upregulated in W0-II, while other positive cell cycle regulators *CCNA2* [21] and *CCNB1/2* [22] were significantly upregulated in W0-III (Figure A1a). A notable increase in mitochondrial read fractions [23] and results from a Gene Ontology enrichment analysis [24] suggested that cells within W0-IV undergo apoptosis (Figure A1b,c).

Subsequently, we analyzed the W0–W9 data using Seurat and conducted a principal component analysis (PCA) based on 5000 highly variable genes, followed by a nonlinear dimensionality reduction using a diffusion map (Figure 2d, left). Thereafter, using MERLoT (v0.2.2) [25], a pseudotime analysis software, we smoothed the gene expression profiles and clustered the cells into five classes, C1–C5 (Figure 2d, right). We observed that cells in W0, W3–W6, and W9 predominantly belong to C1, C2–C3, and C4, respectively (Figure 2e, left). We noticed that the population ratio in the G2/M phase drastically decreased from C1 to C2 but gradually increased from C2 to C4 (Figure 2e, right). Since tamoxifen treatment induces G1 arrest, followed by gradual cell growth recovery [11], we assumed that the C1–C2–C3–C4 axis approximately corresponds to the progression of tamoxifen resistance acquisition, whereas C5 predominantly consists of cells in the S and G2/M phases.

Then, we defined pseudotime *t* for all cells, where we set *t* = 0 and *t* = 1 at the putative start and end points in C1 and C4, respectively (Figure 2f and Appendix C). Consistent with our interpretation of pseudotime as the progression of tamoxifen resistance acquisition, we annotated W0-I and W0-II as the potential tamoxifen-sensitive and resistant populations, respectively (Figure A1d). According to the pseudotime gene expression profiles (Figure 2g), we observed that *ESR1* was downregulated for *t* > 0.5, suggesting that ESR1-independent mechanisms occur after tamoxifen treatment, as previously indicated [11]. We also noticed that key regulators in the cAMP (such as *GREB1*, *XBP1*, and *CREB1*), PI3K-AKT (*RPS6KB1*), RHOA-GTP (*RHOA*), NF-κB (*NFKB1*), and NOTCH (*NOTCH2*) pathways were sequentially upregulated, suggesting the existence of transcriptional relay.

Finally, we investigated the presence of a common transcriptional relay mechanism among different cell lines using external data. In a previous study, preadapted (PA) subpopulations were identified in both MCF-7 and T47D cells [9], a luminal A subtype breast cancer model distinguished from MCF-7 by TP53 mutation. PA cells were characterized by a high CD44 expression and transcriptomic signatures associated with cell survival and EMT in response to estradiol (E2) deprivation. Analyzing the scRNA-seq data of T47D- and T47D-derived long-term estrogen-deprived cells, we discovered a potential common transcriptional relay mechanism underlying the responses to tamoxifen treatment in MCF-7 cells and E2 depletion in T47D cells (Appendix D).

### 2.3. Modeling Pseudotime Gene Expression Patterns

As extensively reviewed, several computational tools have been developed to construct gene regulatory networks from scRNA-seq data [26]. One common approach is transcription factor networks based on the assumption that mRNA levels correlate well with protein levels [27,28]. However, questions arise from the intricate processes, including environmental changes (e.g., drug concentration [29]), the interconnection of signaling pathways, chemical modification, and the nuclear translocation of transcription factors, as well as co-operation with coactivators and corepressors [7]. These processes can significantly influence target gene expressions. To address such complexity, we modeled individual gene expressions using an autoregulatory gene circuit widely used for modeling various biochemical processes (Figure 3a) [30,31], which is exemplified by an ER-related signaling pathway (Figure 3b) [32,33,34,35]. This model operates under the assumptions that all genes are controlled by positive and negative feedback mechanisms, and the strength of this feedback control is determined by the mRNA levels and interactions among intracellular signals. The model is described by the following nonautonomous ordinary differential equation:(1)$$\frac{dx_{i}}{dt}=F\left(x_{i},\;t\right),\;\;\;i=1,\;2,\cdots ,\;I,$$
(2)$$F\left(x_{i},\;t\right)≔a_{i}\left(\frac{{x_{i}}^{k_{i}}}{{x_{i}}^{k_{i}}+K_{i}}\right)\left(\frac{t^{m_{i}}}{t^{m_{i}}+M_{i}}\right)-b_{i}\left(\frac{{x_{i}}^{l_{i}}}{{x_{i}}^{l_{i}}+L_{i}}\right)\left(\frac{t^{n_{i}}}{t^{n_{i}}+N_{i}}\right)+c_{i}-d_{i}x_{i},$$
where *I* is the number of genes, $x_{i}$ is a normalized expression level ($x\geq 10^{-4}$) for gene i, $a_{i}$ and $b_{i}$ are the magnitudes of positive and negative feedback regulation, $c_{i}$ and $d_{i}$ are the transcription and degradation rates, $K_{i}$, $L_{i}$, $M_{i}$, and $N_{i}$ are the half-saturation constants, and $k_{i}$, $l_{i}$, $m_{i}$, and $n_{i}$ are Hill coefficients. All the parameter values are non-negative real numbers.

We separately estimated the model parameters for 6082 genes, which passed a filter removing low-expressed genes—for the C1, C3, and C4 subpopulations—wherein we neglected C2 and C5 due to the lack of sufficient pseudotime lengths (Appendix E and Appendix F). Numerical data fitting in mechanistic models with tens to hundreds of unknown parameters remains challenging [36]. Nevertheless, we addressed this problem using a simplified model (Equation (1)). Here, we used a multi-starts method [37], one of the successful approaches to avoid a suboptimal estimation using different initial guesses, followed by a method using smooth and match estimators [38,39], a fast optimization method fitting a curve $\left(x_{i},t_{i},F(x_{i},t)\right)$ to a given batch of data (Figure 3c). Furthermore, we applied our original *l*_2_-norm penalty method—a parameter set minimizing ${\left\|\left(a_{i},b_{i},c_{i},d_{i}\right)\right\|}_{2}$ is likely to be selected—by assuming that these parameters represent the energy costs of activation and inhibition for gene expressions. This restriction improved the robustness of the estimation.

We observed that fitting errors were generally small (<0.1) for most genes, indicating a good fit of our model to the pseudotime gene expression data (Figure 3d). However, suboptimal parameter sets, derived from different initial guesses during local optimization, also yielded small fitting errors but resulted in different predictions (Figure A2). This suggests the complexity of our model, which may involve an excessive number of parameters compared with the pseudotime data and a possibility of overfitting. Thus, we evaluated the robustness of parameter optimization against small differences in gene expression profiles, controlling the number of representative pseudotime points, denoted as $n_{pc}$ (i.e., the number of “nodes” in Figure 2d, right), which impacts the pseudotime gene expression profiles [25].

To test whether gene-wide median parameter values were consistent across different values of $n_{pc}$, we independently conducted a nonparametric one-way ANOVA for C1, C3, and C4 (Figure 3e). Notably, all 12 parameters for C1 showed no significant differences between $n_{pc}=55,\;56,\;\cdots ,\;60$, whereas parameters $a_{i}$, $b_{i}$, and $N_{i}$ exhibited significant differences between $n_{pc}=55,\;56,\;\cdots ,\;65$ (Figure 3e, left), which may be attributed to fluctuations in parameter values observed for certain genes (Figure 3e, right). Robustness was also observed for C3 and C4 (Figure A3 and Appendix G). Moreover, we verified that all 12 parameters were essential for our model, confirming that every principal component computed from all the estimated parameter values has an almost equal number of contributions (Figure A4). Consequently, we demonstrated the statistically accurate and robust parameter estimation, offering a reliable method for predicting cellular transition states.

### 2.4. Transient and Bifurcation Analysis for Key Regulatory Genes

We conducted a steady-state analysis to identify genes exhibiting multistable expression states at equilibrium. We counted the number of stable fixed points by deriving the steady-state equation from Equation (1) and computing its zero points with a negative differential coefficient. Our analysis revealed that 664, 558, and 680 genes exhibited multistability in the C1, C3, and C4 subpopulations, respectively (Figure 4a). Notably, *RPS6KB1*, a predictor of poor prognosis, cell survival, and growth in ER-positive breast cancers [35], exhibited bistability in both C1 and C4, implying that its low and high expression levels are regulated by a binary switch governing distinct cell fates.

To further investigate the transient gene expression dynamics dependent on the initial conditions, we computed a potential landscape analogous to Waddington’s landscape [40]. This landscape visually represents how gene expression states evolve over time, analogous to marbles rolling on hills and valleys. Derived from Equation (1), the potential function $U\left(x_{i},t\right)$ is given by the following gradient form of the landscape:(3)$$\frac{dx_{i}}{dt}=-\nabla U\left(x_{i},t\right),\;\;\;i=1,\;2,\cdots ,\;I,$$
(4)Uxi, t≔aixiki+1Kiki+1F121, 1+1ki, 2+1ki;−xikiKitmitmi+Mi−bixili+1Lili+1F121, 1+1li, 2+1li;−xiliLitnitni+Ni+cixi−di2xi2,
where _2_*F*_1_ is the Gauss hypergeometric function [41]. Note that the direct differentiation of Equation (4) using the formula ∂zF12α, β, β+1;z=(β/z)[1−z−α−F12α, β, β+1;z] leads to Equation (2). Our observations indicated that the long-time behavior of *RPS6KB1* expression is predominantly determined early in the pseudotime in C1, regardless of initial perturbations (Figure 4b, left). Furthermore, the potential landscape drove most initial states toward higher expression levels in both C1 and C4 (Figure 4b, right), suggesting challenges in controlling *RPS6KB1* expression post cell fate determination.

To interpret the association of *RPS6KB1* with tamoxifen sensitivity and resistance, we partitioned C1 based on normalized expression levels <1 or ≥1. Tamoxifen-sensitive cells in the W0-I subpopulation predominantly belonged to the *RPS6KB1*-low population, while tamoxifen-resistant and growing cells in W0-II and W0-III primarily belonged to the *RPS6KB1*-high population (Figure 4c). Thus, the *RPS6KB1* expression serves as a potential indicator of tamoxifen-resistant states regulated by binary switching pathways.

In a bifurcation analysis, we investigated the relationship between equilibrium expression states and parameters $K_{i}$ and $L_{i}$, which represent the inhibitory effects on positive and negative regulation, respectively. The bistable region for *RPS6KB1* was quite asymmetric with respect to $K_{i}$ and $L_{i}$: Changes in $K_{i}$ were more efficient than those in $L_{i}$ for driving the bistable state into a low-expression state (Figure 4d). This suggests that positive regulation inhibitors effectively control *RPS6KB1* expression, offering potential targets for drug resistance interventions.

### 2.5. Targeting Key Regulatory Genes in Survival and Metastasis-Related Pathways

To comprehensively identify key regulatory genes in the C1 subpopulation, we computed bifurcation diagrams for the genes exhibiting mono- and multistable expression states involved in the cAMP, PI3K-AKT-mTOR, MAPK, NOTCH, and several other pathways, with $K_{i}$ and $L_{i}$ serving as parameters controlling the bifurcation (Figure 5a). Notably, we observed significant asymmetry in the bistable regions; for genes such as *RPS6KB1*, *NRAS*, *ITGB1BP1*, and *ARID1A*, $K_{i}$ exhibited a higher efficacy for inducing bifurcation, whereas, for *TBL1XR1*, $L_{i}$ exhibited a higher efficacy. Conversely, for *JUND*, controlling both $K_{i}$ and $L_{i}$ proved efficient for inducing bifurcation. On the other hand, for *MYC*, neither $K_{i}$ nor $L_{i}$ induced multistability.

Additionally, we integrated the genes exhibiting multistable expression states into a schematic representation of cell survival and metastasis-related pathways, focusing on those involved in focal adhesion and EMT (Figure 5b and Figure A6), which was manually constructed based on the results of the pseudotime analysis and previous reports (Appendix H). This comprehensive mapping provides a method for further understanding the complex interplay between signaling pathways and genes for therapeutic intervention. Furthermore, the observed peaks in *RPS6KB1*, *NFKB1*, *RHOA*, and *NOTCH2* expressions (Figure 2g) suggest cell survival, focal adhesion, and EMT processes in C2 (~3 weeks), C3 (~3–6 weeks), and C4 (~3–9 weeks), respectively, adding further information to the dynamic processes of drug resistance acquisition.

## 3. Discussion

We proposed an approach that combines mathematical modeling with a pseudotime analysis using single-cell time-series transcriptome data to identify key regulatory genes involved in drug resistance acquisition in breast cancer. In contrast to previous systems biology models that rely on predefined kinetic networks [42,43,44,45], our approach offers a widely applicable and computationally efficient method for modeling each gene expression. We demonstrated statistically accurate and robust parameter estimation for approximately 6000 genes and identified 560–680 genes exhibiting multistable expression states, including *RPS6KB1*, a predictor of poor prognosis, cell survival, and growth in ER-positive breast cancers [35]. Furthermore, our analysis revealed that the long-time behavior of the *RPS6KB1* expression is predominantly determined early in the pseudotime and can be effectively modulated by inhibitors for positive regulation. Nevertheless, experimental validation is important and potentially involves targeting the drug design, focusing on the key genes.

Despite yielding small fitting errors, suboptimal parameter sets produced divergent gene expression dynamics, indicative of the sensitivity of our model, which may involve an excessive number of parameters. Thus, future work should focus on model parameter reduction, possibly by sharing parameters among specific genes and mapping the original and reduced parameter spaces. A further model reduction will also mitigate the risk of overfitting. Furthermore, cell-to-cell heterogeneity should not be disregarded. While some genes, such as *NFKB1* and *NOTCH2*, exhibited similar pseudotime gene expression patterns (Figure 2g), the Pearson correlation coefficient across cells was <0.05. This low correlation coefficient indicates a considerable variability in the gene expression among cells. Oral cancer cells exhibit heterogeneous responses to transforming growth factor β stimuli via multiple EMT pathways [46]. This emphasizes the critical role of heterogeneous signaling states in affecting the drug response, which should be incorporated into future modeling approaches.

Finally, several limitations are worth noting. We identified potential drug-sensitive and -resistance subpopulations in the tamoxifen-untreated cells and found that genes exhibiting bistability were concentrated in the MAPK-related pathways associated with cell survival (Figure 5b), suggesting that cell survival is regulated by switch-like responding pathways. Additionally, a “pre-adapted” MCF-7 subpopulation exhibiting drug tolerance has been identified [9,10]. Although these findings are useful, our analysis is currently limited to only the published scRNA-seq data [11]. This does not ensure its generalizability to other cancer datasets. Therefore, a further analysis using multiple datasets such as time-series transcriptome [9,10] and proteome data [47] is necessary to improve our data-driven model and thoroughly understand cancer drug resistance acquisition.

## 4. Conclusions

Our modeling approach, combined with a single-cell pseudotime analysis, revealed genes with multistable expression states and elucidated their regulatory mechanisms through a bifurcation analysis. Mapping the key regulatory genes into a molecular network considering cell survival and metastasis-related pathways provided a comprehensive understanding of the interplay between the signaling pathways and genes, serving as a powerful tool for deciphering the complexities of drug resistance mechanisms in breast cancer. Future analyses should aim to uncover additional insights into the mechanisms of cancer drug resistance by leveraging larger datasets.

## Figures and Tables

**Figure 1 cancers-16-01884-f001:**
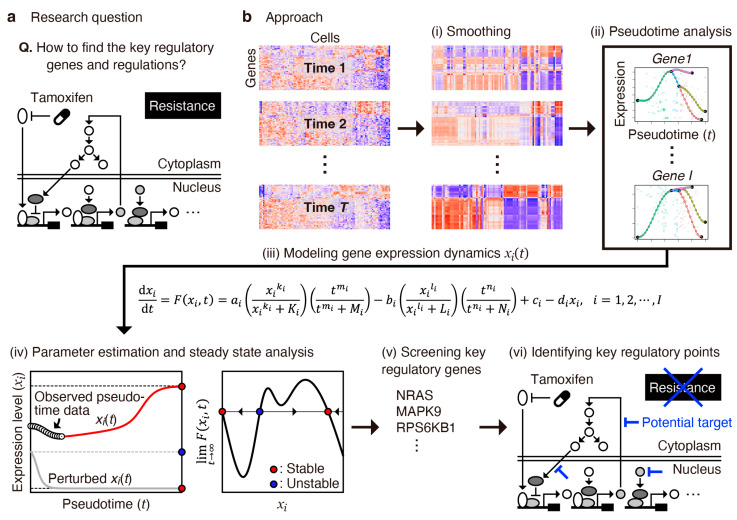
Research question and overview of our modeling approach. (**a**) Research question: How to identify the effective targets regulating complex molecular systems upon drug resistance acquisition. (**b**) Modeling approach: (i) Smoothing of single-cell data; (ii) pseudotime analysis, which outputs pseudotime gene expression data; (iii) mathematical modeling of each gene expression using an ordinary differential equation; (iv) parameter estimation by fitting the model (left) and steady-state analysis (right); (v) screening of key regulatory genes exhibiting multistable expression levels at equilibrium; and (vi) mapping of key regulatory genes into a molecular network associated with drug resistance acquisition and identifying potential regulatory points.

**Figure 2 cancers-16-01884-f002:**
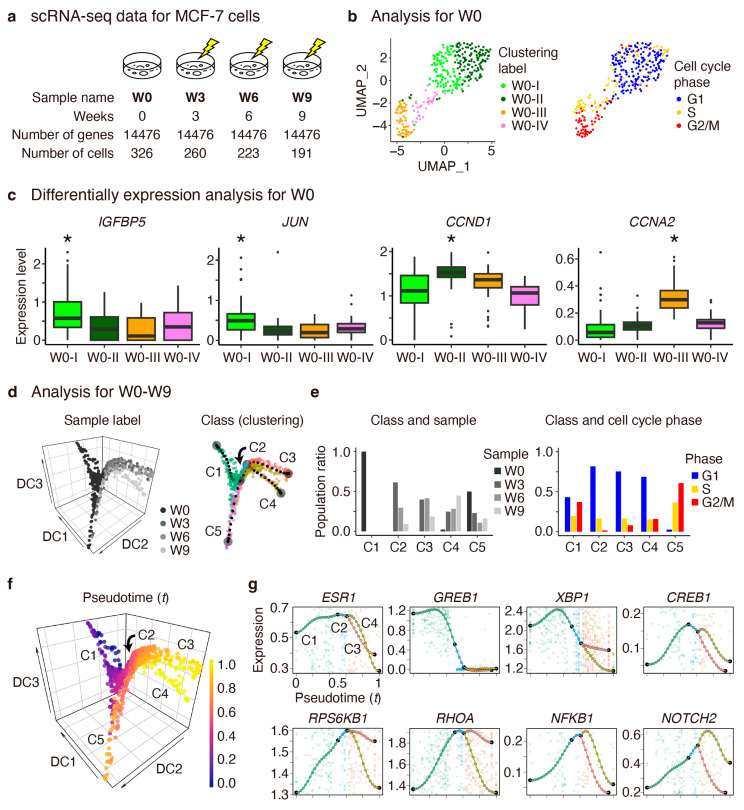
Statistical and pseudotime analysis of time-series single-cell RNA-seq data of MCF-7 cells under continuous tamoxifen treatment. (**a**) Overview of the dataset. (**b**) Uniform manifold approximation and projection (UMAP) plots for the control sample (W0), showing the clustering results (left) and cell cycle phases (right) computed using Seurat. (**c**) Box plots for W0, showing the normalized expression levels of differentially expressed genes. Adjusted *p*-values for the clusters marked with asterisks indicate statistical significance (* *p* < 0.001, Mann–Whitney U test by Seurat). (**d**) Diffusion map plots for all samples (W0–W9), showing the sample labels (left) and clustering labels (right) computed using MERLoT. (**e**) Population ratios across clustering versus sample labels (left) and clustering versus cell cycle phases (right). (**f**) Diffusion map plot showing pseudotime *t*, where we set *t* = 0 and *t* = 1 at the putative start and end points of branches C1 and C4, respectively. (**g**) Pseudotime gene expression profiles of key regulators in estrogen receptor pathway (such as *ESR1*), cAMP pathway (*GREB1*, *XBP1*, and *CREB1*), PI3K-AKT (*RPS6KB1*), RHOA-GTP (*RHOA*), NF-κB (*NFKB1*), and NOTCH (*NOTCH2*) pathways, which are sequentially upregulated from top left to bottom right, suggesting the existence of transcriptional relay.

**Figure 3 cancers-16-01884-f003:**
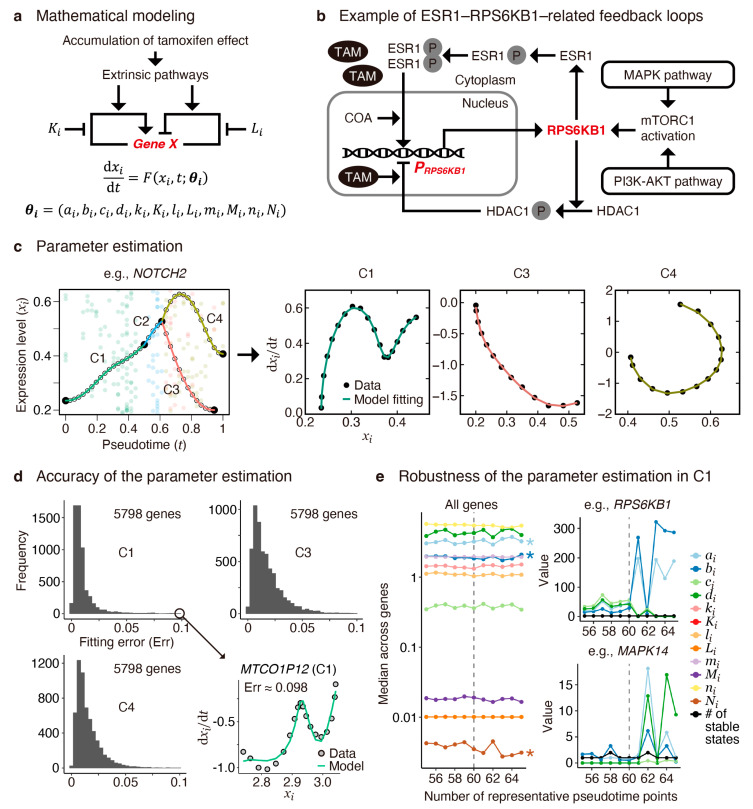
Mathematical modeling, parameter estimation, and evaluation of accuracy and robustness. (**a**) Schematic representation of the mathematical model in which intracellular tamoxifen levels affect gene regulation via extrinsic pathways. The arrows and blunt arrows represent positive and negative regulation, respectively. Particularly, the $K_{i}$ and $L_{i}$ parameters represent inhibitory effects on positive and negative regulation, respectively. (**b**) Schematic representation of *RPS6KB1* regulation, an example of estrogen-receptor-related signaling. (**c**) Demonstration of parameter estimation using *NOTCH2* pseudotime trajectories. Parameters were independently optimized for subpopulations C1, C3, and C4, whereas C2 and C5 were neglected due to insufficient pseudotime length. (**d**) Histograms of the fitting errors showing that the errors are generally small (≤0.1) for most genes (5798 genes out of 6082), as exemplified by the result for *MTCO1P12* in C1 at the right bottom. (**e**) Gene-wide median parameter values for C1 (left), along with individual parameter values and the number of stable fixed points for *RPS6KB1* (top right) and *MAPK14* (bottom right) across different numbers of representative pseudotime points ($n_{pc}$). In this study, $n_{pc}=60$ was used (indicated by vertical broken lines). According to the Kruskal–Wallis test, all 12 parameters showed no significant differences between $n_{pc}=55,\;56,\;\cdots ,\;60$ (*p*-value > $10^{-4}$), whereas $a_{i}$, $b_{i}$, and $N_{i}$ exhibited significant differences between $n_{pc}=55,\;56,\;\cdots ,\;65$: * *p*-values < $10^{-10}$.

**Figure 4 cancers-16-01884-f004:**
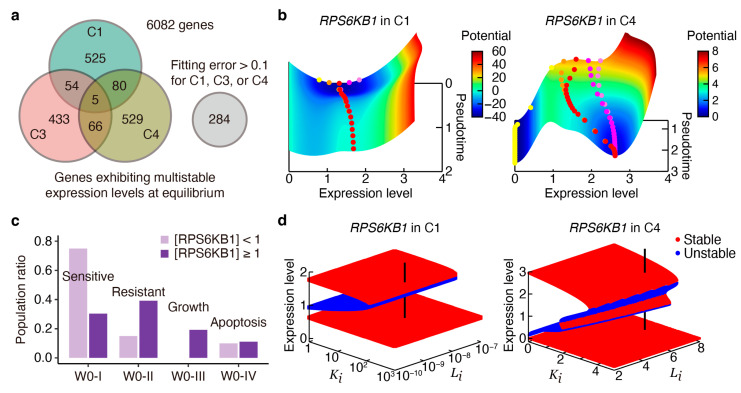
Transient and bifurcation analysis for “multistable genes” exhibiting multistable expression levels at equilibrium. (**a**) The number of multistable genes identified in C1, C3, and C4 subpopulations and those whose pseudotime gene expression profiles could not be well fitted by the model. (**b**) Potential *RPS6KB1* expression landscape in C1 (left) and C4 (right), providing a visual representation of how gene expression states evolve over pseudotime. Colored, filled circles are snapshots of inferred expression dynamics given several initial values: red circles correspond to the pseudotime data, while yellow, orange, magenta, and violet circles correspond to initial perturbations with 0.6-, 0.8-, 1.2-, and 1.4-fold changes, respectively. (**c**) Population ratios across W0-I (tamoxifen-sensitive cells), W0-II (tamoxifen-resistant cells), W0-III (growing cells), and W0-IV (apoptotic cells) versus *RPS6KB1*-low and -high populations. (**d**) Bifurcation diagrams showing stable (red) and unstable (blue) expression states dependent on parameters $K_{i}$ and $L_{i}$, which represent inhibitory effects on positive and negative regulation, respectively. The vertical black lines represent the parameter set estimated from the pseudotime data.

**Figure 5 cancers-16-01884-f005:**
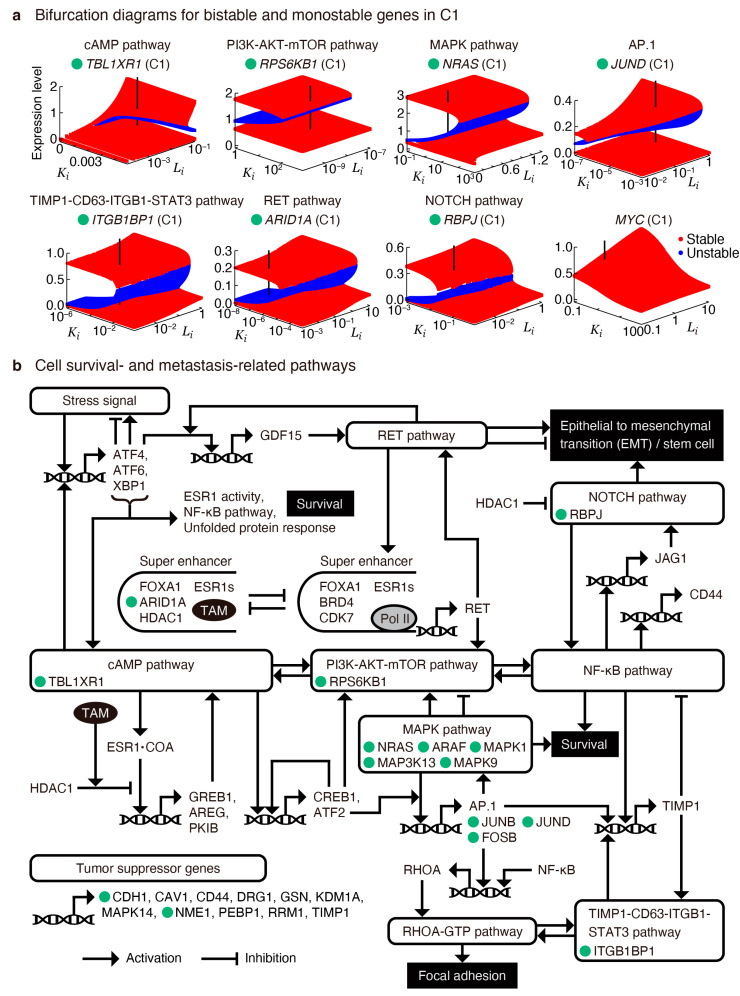
Integration of key regulatory genes into cell survival and metastasis-related pathways involved in focal adhesion and epithelial–mesenchymal transition (EMT). (**a**) Bifurcation diagrams for genes exhibiting bistable (*TBL1XR1*, *RPS6KB1*, *NRAS*, *JUND*, *ITGB1BP1*, *ARID1A*, and *RBPJ*) and monostable (*MYC*) expressions in C1 population that are involved in cAMP, PI3K-AKT-mTOR, MAPK, AP.1, TIMP-1-CD63-ITGB1-STAT3, RET, and NOTCH pathways. (**b**) Comprehensive mapping of the key regulatory genes, marked by green-filled circles, into a schematic representation of cell survival and metastasis-related pathways, focusing on those involved in focal adhesion and EMT. The green-labeled genes are shown in (**a**). TAM, tamoxifen.

## Data Availability

The scRNA-seq data of MCF-7 cells treated with tamoxifen are available in the Gene Expression Omnibus with accession code DRA009126. The scRNA-seq data of T47D- and T47D-derived long-term estrogen-deprived cells were obtained from the Gene Expression Omnibus with accession code GSE122743. The codes for analyzing data in this article are available on Github (https://github.com/keita-iida/PSEUDOTIMEABC, accessed on 27 April 2024).

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
