# Peer review of "Identifying Key Regulatory Genes in Drug Resistance Acquisition: Modeling Pseudotime Trajectories of Breast Cancer Single-Cell Transcriptome"

_cancers, 2024, doi:10.3390/cancers16101884_

Round 1

Reviewer 1 Report

Comments and Suggestions for Authors

I. Review report to the Authors:

Summary / significance: The role of using biomarkers for analysing BrCa not only on classical morphological but also on omics level is becoming increasingly important, especially when it comes to personalized medicine approaches. This paper by Iida & Okada presents a comprehensive bioinformatic analysis that enhances the understanding of drug resistance mechanisms in breast cancer. The integration of mathematical modeling with pseudotime analysis using single-cell transcriptome data is particularly noteworthy. This approach effectively identifies key regulatory genes involved in drug resistance acquisition, shedding light on the complex interplay between signaling pathways and genes.

Level of interest/merit: Understanding resistance features of cancer is key for the identification of novel powerful markers and therapeutic tools. This is an interesting computational study focussing on analysis of the predictive power of BrCa markers in clinical context.

Comment: The information provided in this article is extremely high and comprehensive, also summarized in instructive Figures. The level of computational background is very high. The detailed dissection of cellular heterogeneity and definition of pseudotime for MCF-7 cells provide insights into transcriptional regulation during drug resistance acquisition. The modeling of pseudotime gene expression patterns, coupled with parameter estimation for thousands of genes, demonstrates a rigorous computational analysis that yields statistically accurate and robust results. The transient and bifurcation analysis for key regulatory genes, especially the identification of multistable expression states and their implications for cell fate determination, makes a significant contribution. Additionally, the mapping of key regulatory genes into survival and metastasis-related pathways offers a comprehensive understanding of the regulatory mechanisms.

Overall, the bioinformatic analysis presented in the paper leaves no gaps, and it is evident that all necessary steps have been taken. There is nothing further to request in terms of bioinformatics.

Specific comments:

1) Manuscript header could be more explicit, mentioning at least breast cancer.

2) The authors could have highlighted an interesting new resistance-associated gene or pathway.

Author Response

We appreciate for reviewing our manuscript and your kind comments. Please see the attachment word file.

Reviewer 2 Report

Comments and Suggestions for Authors

cancers-3006677

The Manuscript " Identifying key regulatory genes in drug resistance acquisition: Modeling pseudotime trajectories of single-cell transcriptome" by the Authors Keita Iida and Mariko Okada 

In this Article the authors carefully drafted this manuscript disclosed their results and they identify distinct subpopulations within tamoxifen-treated breast cancer cells, revealing heterogeneity in drug response. They have identified approximately 560–680 genes exhibiting multistable expression states sheds in each subpopulation. Their study identifies key regulatory genes, like RPS6KB1, which is associated with cell survival and drug resistance acquisition in estrogen receptor-positive breast cancer cells which provides valuable insights into potential therapeutic targets.

Authors have carefully drafted this manuscript with all supporting information and provided sufficient citation in the manuscript.

The manuscripts can be published, and it will be useful from scientific readers point of view. However, I have few comments. 

Authors can consider rewriting in 2.2. first paragraph

Authors can consider checking figures such as Fig.2 (b)

Author Response

(The authors gave the same response as above.)

Reviewer 3 Report

Comments and Suggestions for Authors

The manuscript “Identifying key regulatory genes in drug resistance acquisition: Modeling pseudotime trajectories of single-cell transcriptome” is devoted to transcriptomic analysis of breast cancer cells to understand the mechanisms of drug resistance. To achieve this, the authors have done a great deal of work to create mathematical models of such resistance and determined the expression spectra of more than 600 genes responsible for the formation of drug resistance. Mathematical processing, taking into account various variants of data errors, multiple duplication and testing of the created models made it possible to create a predictive mechanism for assessing the profiles of cell resistance to chemotherapeutic drugs. Moreover, this work characterizes in detail the expression profiles of more than 600 genes and signaling pathways that are involved in drug resistance profiles.

The work is very relevant, performed at a very high level, and written competently. The mathematical models are also described in great detail. I believe that the work can be published in the scientific journal Cancer in the form in which it was presented to the reviewer.

Author Response

We appreciate for reviewing our manuscript and your kind comments. Following the comments from Reviewers 1 and 2, we have revised our manuscript. Please find the detailed in track changes in the re-submitted files.

Reviewer 4 Report

Comments and Suggestions for Authors

The authors performed mathematical modeling using pseudotime analysis of single-cell RNA sequencing done on MCF-7 cell line treated with tamoxifen. The aim of analysis was to understand the mechanisms connected with cancer drug resistance or sensitivity which is extremely important in modern cancer treatment. The analysis was performed on 6082 genes and mathematical modeling approach was used to find genes which modified the response to treatment. They found RPS6KB1 gene which is predictor of poor prognosis and they also found potential drug-sensitive or resistant subpopulations of cells which were not treated with tamoxifen. Their finding shows that it should be possible to find out whether the patient will respond or not to treatment. The authors are critical about their finding that it needs further analysis. The manuscript is interesting since it presents the multigene analysis together with the signaling pathways evaluation.

Author Response

(The authors gave the same response as above.)
